# Evaluation of the Bioactive Compounds, and Physicochemical and Sensory Properties of Gluten-Free Muffins Enriched with Persimmon ‘Rojo Brillante’ Flour

**DOI:** 10.3390/foods11213357

**Published:** 2022-10-25

**Authors:** Sepideh Hosseininejad, Virginia Larrea, Gemma Moraga, Isabel Hernando

**Affiliations:** Department of Food Science and Technology, Universitat Politècnica de València, Camino de Vera s/n, 46021 Valencia, Spain

**Keywords:** bakery products, celiac disease, *Diospyros kaki*, tannins, carotenoids, antioxidant activity, recovery index

## Abstract

Because of the nutritional need of patients with celiac disease, producing quality gluten-free diet food is important. This study evaluated the use of persimmon flour on the properties of muffins. Persimmon flour obtained from the astringent variety ‘Rojo Brillante’, which is often discarded due to its characteristic astringency, was added to muffins replacing corn flour (10%, 20%, and 30%). Despite the height differences between the control muffin and the muffins with persimmon flour, similar mechanical parameters were obtained. As the percentage of persimmon flour increased, the muffin color was darker, turning toward a more reddish hue, mainly because of the intensification of nonenzymatic browning reactions. The sensory results showed high scores for taste attributes, the texture attributes were similar to the control, and astringency was hardly detected when persimmon flour was used. The content of tannins and carotenoids and their antioxidant activity increased significantly with an increasing amount of persimmon flour. After in vitro digestion, high recovery index values of soluble tannins and carotenoids were obtained in muffins with added persimmon flour. This study shows that the use of persimmon flour as a functional ingredient offers an opportunity to develop quality gluten-free muffins that reduce agricultural losses. Astringent varieties can be used, without applying a de-astringency treatment, as the astringency is removed during muffin baking due to tannins’ insolubilization.

## 1. Introduction

Considering the increasing importance of binomial nutrition and health, some functional ingredients can be added to daily foods to form an easy-to-deliver mechanism to enhance consumer health. Many studies have been conducted in the past on the development of bakery products containing dietary fiber from plant sources, such as fruits, vegetables, and their by-products. Nath et al. [1] investigated the effect of adding 2–10% red capsicum pomace powder to wheat flour on the physical characteristics, bioactive components, and antioxidant activities of muffins. The partial substitution of 5%, 7%, and 9% of whole wheat flour by mutamba fruit (*Guazuma ulmifolia* LAM.) flour in bread was addressed by Assis et al. [2]. Sahni et al. [3], in their review, compiled studies about the use of different fruit and vegetable pomace, such as apple, carrot, orange, or beetroot, to prepare bakery products.

Bakery products are an important category of consumer meals. They have been widely consumed by people of all ages due to their taste, flavor, appearance, and soft texture. Muffins are sweet, high-calorie bakery products, favored for their compact size, shape, and portability [4]. However, despite the acceptance of the consumer, the nutritional profile can be improved, especially when it is destined for specific populations with chronic diseases, such as celiac disease. Celiac disease is an autoimmune disorder that primarily affects the small intestine caused by a permanent intolerance to gluten—a group of proteins present in cereals such as wheat, barley, rye, or oat [5,6]. Its consumption in small amounts produces villus atrophy of the intestinal mucosa that leads to the altered absorption of nutrients [7,8].

Preparing gluten-free muffins is difficult because of the typical porous structure and the high volume of the muffins, which give them a spongy texture. To obtain such a final structure, a stable batter containing small bubbles is required [9], and gluten helps to retain these bubbles. One of the main challenges in obtaining high-quality gluten-free products is finding structuring agents to simulate gluten, such as egg protein [10] or hydrocolloids [11,12]. Today, scientific data on gluten-free muffins made with different types of flour are available, such as chickpea [13], rice [14], corn [15], or buckwheat [16].

Persimmon is a delicious fruit with a high content of fiber, minerals, vitamins, phenolic compounds, and carotenoids, with a wide range of biological activities and beneficial effects against diseases such as oxidative stress, cancer, hypertension, diabetes mellitus, and atherosclerosis [17,18,19]. The carotenoids are responsible for color and antioxidant capacity [20,21]. Among the phenolic compounds, persimmon is rich in tannins, which are responsible for astringency. In this regard, persimmon varieties can be divided into two categories according to astringency: astringent (with a higher soluble tannin content) and non-astringent (lower soluble tannin content) cultivars. The astringent varieties, such as ‘Rojo Brillante’, need to pass through a de-astringency treatment to be consumed; otherwise, they will be discarded for consumption and will become part of agri-food industry losses [20]. Moreover, the high volumes of production of ‘Rojo Brillante’ persimmon, the short commercial season of this cultivar (between mid-October and the end of December), the strict quality control to which the fruit is subjected, and the inefficiency of de-astringency treatment in advanced ripening stages produce extra surpluses [19].

Therefore, one of the current challenges for the persimmon producers is to seek strategies to increase the value of discarded fruits, developing persimmon-derived foods. In this context, the addition of persimmon flour to muffins provides a unique opportunity to develop innovative fusion products with an improved nutritional profile, reducing agri-food losses. To the best of our knowledge, little background information on muffins enriched with persimmon flour is available. Therefore, this study was conducted to evaluate the effect of the addition of persimmon flour on the physicochemical and sensory properties of gluten-free muffins. Furthermore, the stability of tannins, carotenoids, and their antioxidant activity during in vitro gastrointestinal digestion was also determined. The main novelty of this work is to use an astringent persimmon flour to obtain a non-astringent food product, avoiding the application of a de-astringency treatment, which involves an extra cost.

## 2. Materials and Methods

### 2.1. Persimmon Flour Preparation

Persimmon fruit (*Diospyros kaki* Thunb. cv. Rojo Brillante), without applying any de-astringency treatment, was provided by Instituto Valenciano de Investigaciones Agrarias (IVIA, Valencia, Spain). These persimmon fruits were harvested from a local grove of L’Alcudia (Valencia, Spain) in early December, corresponding to the ripening V [22] with a characteristic orange color. The fruits were washed and cut transversally into 5 mm thick slices according to González et al. [23]. Freeze-dried persimmon was ground in a grinder mill (Molinex, Indonesia) to obtain the persimmon flour. The amounts of soluble tannins content, ferric reducing antioxidant power (FRAP), and carotenoids in persimmon flour were 1.14 ± 0.15 g GAE/100 g, 108.52 ± 10.39 µmol Trolox/g, and 3.41 ± 0.04 mg beta-carotene/100 g, respectively.

### 2.2. Muffins Preparation

Four muffin batter formulations were prepared according to Table 1. The muffin batter ingredients included corn flour (Maizena, Unilever España, S.A, Barcelona, Spain; composition per 100 g: 86 g carbohydrate, 0.5 g protein, and 0.5 g fat); sugar (Pfeifer and Langen GmbH and Co., Cologne, Germany); whole liquid egg (Huevos Guillen S.L., Valencia, Spain); refined sunflower oil (Sovena España S.A., Sevilla, Spain); xanthan gum (Cargill France SAS, Puteaux, France); baking powder (Hacendado, Valencia, Spain). The persimmon flour (composition per 100 g: 84 g carbohydrate, 3.7 g protein, and 1.6 g fat) was added to the batter replacing 0% ((control) F0), 10% (F1), 20% (F2), and 30% (F3) corn flour. The preparation of muffins was conducted according to Martínez-Cervera et al. [9].

### 2.3. Color Measurements

The color of the muffin, crumb, and crust was measured using a CR-400 colorimeter (Minolta Co., Ltd., Osaka, Japan). The result was expressed in the CIE L*a*b* system, using the standard light source C and the standard visual angle of 2°.

The total color difference (ΔE*) between muffins enriched with persimmon flour and the control sample was calculated using Equation (1) [24].
(1)$$∆E^{*}=[{\left(∆L^{*}\right)}^{2}+{\left(∆a^{*}\right)}^{2}+{\left(∆b^{*}{)}^{2}\right]}^{1/2}$$

### 2.4. Muffin Height and Crumb Structure

The muffins were cut vertically and scanned following the method described by Diez-Sánchez et al. [25]. The muffin height and three macroscopic characteristics of the structure of the crumb cell were calculated, namely the air cell count (number of cells per field (30 × 30 mm)), total area of the cell, and the size of the air cell. Data were obtained from five images for each formulation.

### 2.5. Texture Measurements

Texture profile analysis (TPA) was conducted according to Diez-Sánchez et al. [26]. A 35 mm diameter aluminum plate (P/35) was used. Eight repetitions were performed on each sample.

### 2.6. Sensory Analysis

A descriptive sensory analysis was performed using a standard quantitative descriptive analysis technique (QDA). A sensory panel composed of 10 trained judges with experience in assessing muffin quality were used. A consensus vocabulary of 10 descriptors was developed to characterize the samples: Sweetness, Fruity taste, Caramel taste, Astringency, Hardness, Sponginess, Chewiness, Moist texture, Puffy appearance, and Dark color (in the crumb). The four muffins F0, F1, F2, and F3 were presented in random order and coded with random three-digit numbers, and a 10 cm structured line scale was used for scoring. A sensory profilogram was generated using mean values.

### 2.7. Soluble Tannin Content (STC)

The STC was determined using the Folin–Ciocalteu colorimetric method described by Arnal et al. [27]. The results were expressed as g of gallic acid equivalent (GAE)/100 g sample. Tannin extraction was performed in triplicate.

### 2.8. Total Carotenoid Content (TCC)

TCC was determined according to Nath et al. [1]. The results were expressed as mg β-carotene/100 g of the sample. Carotenoid extraction was performed in triplicate.

### 2.9. Antioxidant Activity (FRAP and DPPH)

Antioxidant activity (FRAP and DPPH) was determined on tannin and carotenoid extracts according to the method of Benzie et al. [28] and Matsumura et al. [29], respectively. A standard curve was performed using Trolox as standard and the results were expressed as µmol Trolox/g.

### 2.10. In Vitro Digestion

An in vitro gastrointestinal tract model was used to simulate the biological fate of ingested samples, according to the method of Brodkorb et al. [30]. Three stages were simulated: oral, gastric, and small intestine. All enzymes used in the analysis were supplied by Sigma-Aldrich (Spain).

The digestion process was conducted in a ‘Carousel 6 Plus’ reaction station (Radleys, UK) under controlled temperature (37 °C), agitation (150 rpm), without light, and under an N_2_ atmosphere according to González et al. [31]. To analyze the effect of in vitro digestion on the content of soluble tannins and carotenoid, the recovery index was calculated. By comparing with the amount in the undigested sample, the recovery index gives the amount recovered after intestinal digestion [32]. The recovery index (RI) was measured according to Equation (2).
(2)$$Recovery\;index\;\left(\%\right)=\frac{DF}{UDF}\times 100$$

*DF* (digested fraction) is the content of bioactive compounds in the digested fraction after the intestinal digestion phase and *UDF* (undigested fraction) is the content of bioactive compounds quantified in fresh muffins.

### 2.11. Statistical Analysis

Data were analyzed by analysis of variance (ANOVA) using the least significant difference test (LSD) with a 95% confidence interval to compare test means (Stat graphics Centurion XVII Manugistics, Inc., Rockville, MA, USA).

## 3. Results and Discussion

### 3.1. Color Measurements

Table 2 indicates the color attributes of the crumb and crust of the different muffins. The control muffin showed a golden-brown crust and a yellow crumb, which are characteristic of bakery products.

As the percentage of persimmon flour increased, the muffin was darker, with the muffin crust and crumb turning to red hues and losing some of its yellowness. Although darker colors are undesirable in numerous foods, consumers believe that darker muffins are healthier than lighter muffins [33]. In this study, the higher the percentage of persimmon flour added, the darker the result, leading to a significant (*p <* 0.05) decrease in L* values.

Consistent with our results, Abdallah et al. [34] reported a decrease in the L* values of wheat flour cupcakes when persimmon puree was added. Yesilkanat and Savlak [35] also reported a decrease in L* values in gluten-free cakes enriched with persimmon powder [35].

The effect of persimmon flour on C* and h* was significant (*p <* 0.05) on both the muffin crust and crumb, turning the muffin color less saturated and with a more reddish hue angle. The differences in color exhibited by the different formulations can mainly be attributed to the intensification of nonenzymatic browning (caramelization and Maillard reaction) that occurred as the amount of incorporated persimmon flour increased [36], but also could be related to the presence of carotenoids, such as β-cryptoxanthin and β-carotene, responsible for the reddish-orange color of the persimmon fruit [35]. ΔE* values, ranging from 20.91 to 44.07, were higher in the crumb. As expected, the higher the percentage of persimmon flour, the higher ΔE* values were observed.

### 3.2. Muffin Height and Crumb Structure

Figure 1 indicates the effect of persimmon flour on the macroscopic characteristics of the crumb structure. The left column corresponds to the scanned crumbs of the muffins and the right column to their binarized images. Visual analysis shows a crumb with larger air cells in F0 and F1 compared to F2 and F3 muffins, where the air cells are smaller and more numerous.

The quantified parameters air cell count, total cell area, air cell size, and height of the muffin are shown in Table 3. F0 and F1 had a significantly lower (*p <* 0.05) count of air cells and a significantly increased (*p <* 0.05) value of air cell size. The addition of persimmon flour (F2 and F3) led to an increase in the number of cells, but the air cell had a smaller size. Consequently, a similar amount of air was incorporated in the muffins structure, and no significant differences (*p >* 0.05) in total cell area were found among the samples. Manaf et al. [37] also reported an increase in the count of air cells in muffins when avocado puree was incorporated.

A higher height was obtained in the control muffin, without significant differences (*p >* 0.05) between F1, F2, and F3. This agrees with the results obtained by Nath et al. [1] who reported a slight decrease in muffin height after fortification with red capsicum pomace powder (CP). Baixauli et al. [38] also found a decrease in muffin height when the flour was replaced by resistant starch.

### 3.3. Texture Measurements

Table 4 indicates the values of the TPA parameters hardness, springiness, cohesiveness, chewiness, and resilience obtained in the different muffins. Despite the differences observed in muffin height and crumb structure, similar mechanical parameters were obtained in all muffins.

Significant differences (*p <* 0.05) in any mechanical parameter were found neither between F0 and F2 nor between F2 and F3. However, F1 had the lowest hardness and chewiness value. Furthermore, F3 showed a greater cohesiveness than F0 and greater resilience than F0 and F1.

No significant differences in springiness were found between the samples, which could be related to the same values of the total aerated area.

In contrast to our results, Yesilkanat and Savlak [35] reported that hardness and chewiness increased, but springiness, cohesiveness, and resilience decreased when persimmon powder was added. They used persimmon powder, obtained from the astringent Hachiya variety, as a partial sugar substitution in gluten-free cake formulated with rice flour. The different variety of persimmons and the different batter formulation used to make the muffins could be the reasons for the different results obtained in the mechanical properties.

### 3.4. Sensory Analysis

The results of the descriptive sensory analysis performed on the four muffins are shown in Figure 2.

No significant (*p >* 0.05) differences were found in the chewiness, moist texture, and hardness attributes of the four muffins, which agree with the instrumental texture measurements. However, F1 and F2 were significantly (*p <* 0.05) spongier than F0. Regarding taste attributes, the addition of persimmon flour produced a significant increase (*p <* 0.05) in sweetness, fruity taste, and caramel taste. However, no differences (*p >* 0.05) were found between F1, F2, and F3 for sweetness. In this sense, small amounts of persimmon flour led to an increase in sweetness detectable by the panel.

A significant increase (*p* < 0.05) was observed in astringency when comparing F2 and F3 with F1 and F0. The panel did not detect the addition of 10% persimmon flour to the muffins, as no significant differences (*p* > 0.05) were found for this attribute between the control and F1. However, it is important to remark that all the muffins, even those with a 30% replacement of corn flour with astringent persimmon flour, received low scores in the astringency attribute, with all being below 2.44.

When increasing the amount of persimmon flour, significant differences) *p <* 0.05) in dark color were observed, and the score increased from 0.61 for the control sample to 8.63 for the F3 sample, which is consistent with instrumental color measurements. The puffy appearance was significantly reduced (*p <* 0.05) in F2 and F3, from 8.02 in the control F0 to 3.96 and 3.23 in F2 and F3, respectively.

Other researchers have addressed the effect of using persimmon in the sensory properties of bakery products. Yesilkanat and Savlak [35] investigated the effects of the partial substitution of sugar with persimmon powder (0%, 20%, 40%, 60%, and 80%) in a gluten-free cake formulation. They reported that the substitution of persimmon powder allowed sensory-acceptable cakes. Substitution levels up to 40% lead to cakes statistically similar to the controls (*p >* 0.05) of crumb and crust color, odor, taste, hardness, and overall acceptability on a 7-point hedonic scale. Abdallah et al. [34] studied the use of persimmon puree as a sugar substitute (33.3, 50, 66.6, and 83.3% substitution) for a cupcake formulation. They reported that the substitution of 50–83.3% sugar for persimmon puree resulted in lower sensorial scores compared to controls, whereas the replacement of 33.3% sugar led to an increase in taste, flavor, color, texture, and overall acceptability scores.

### 3.5. Soluble Tannin Content and Antioxidant Activity

The STC and the antioxidant activity of the soluble fraction for the different muffins are presented in Table 5. Persimmon flour provided a significant increase (*p <* 0.05) in the content of these bioactive compounds; the higher the amount of persimmon flour that was added, the higher the STC that was obtained.

It should be noted that the STC in the astringent persimmon flour used for the elaboration of muffins was 1.14 ± 0.15 g GAE/100 g, but this value was reduced mainly because tannin insolubilization can occur during baking. This is the main finding of our study, and the results are in accordance with the low scores on the astringency attribute obtained in the sensory analysis. González et al. [23] reported that drying significantly reduces the content of soluble tannins. According to their study on dried persimmon snacks, fresh astringent samples showed a decrease in STC after drying at 40 and 60 °C. Hot air-drying treatments typically reduce the content of soluble tannins due to the transformation of soluble tannins into insoluble tannins. Previous studies have found that when samples are dried, they lose bioactive compounds; these losses have been attributed not only to the thermal degradation of phytochemicals [39] but also to the transformation of soluble forms of tannins into their insoluble forms [40].

The level of antioxidant activity measured by the FRAP and DPPH methods showed a similar trend (Table 5). F3 had the highest levels of antioxidant activity (4.79 and 1.10 µmol Trolox/g, respectively) followed by F2 (3.96 and 0.32 µmol Trolox/g, respectively), whereas control values were 0.52 and 0 µmol Trolox/g, respectively.

Similar to our findings, Yesilkanat and Savlak [35] observed that replacing sugar with persimmon flour (0, 20, 40, 60, and 80%) in gluten-free cakes increased the total phenolic content by over 300% in the 80% substituted flour cake compared to the control cake. In addition, FRAP values increased as the concentration of persimmon powder in the cake formulation increased. According to a study by Nath et al. [1], the addition of CP in the preparation of muffins (0, 2, 4, 6, and 10%) increased the total phenolic content from 26.7 mg GAE/100 g in the control to 37.36 mg GAE/100 g in muffins fortified with 10% CP, showing a 40% increase.

### 3.6. Total Carotenoid Content and Antioxidant Activity

Table 6 indicates the TCC and the correspondent antioxidant activity (FRAP and DPPH method) in the different muffins. TCC significantly (*p* < 0.05) increased in enriched muffins compared to control F0; however, no significant (*p >* 0.05) differences between F2 and F3 were observed.

Muffins enriched with persimmon flour also increased the corresponding antioxidant activity (Table 6), improving the nutritional profile. The results of the FRAP and DPPH method did not show significant (*p >* 0.05) differences between F0 and F1; however, the antioxidant activity was significantly higher (*p <* 0.05) in F2 and F3.

These results are consistent with the results of Sello et al. [41] after adding pumpkin powder (5 and 10%) to cupcakes. According to their findings, fortified cupcakes with 10% pumpkin powder had the highest levels of total carotenoids, total phenols, and antioxidant activity. Campos et al. [42] also observed an increase in TCC after replacing wheat flour with freeze-dried pumpkin powder (0, 10, and 20%) in cake formulations.

### 3.7. Effect of In Vitro Digestion on Soluble Tannin Content, and Carotenoid and Antioxidant Activity

The recovered STC in the digesta of the small intestine, after in vitro digestion, increased significantly for all muffins (Table 7) compared to undigested muffins (Table 5). No significant differences (*p >* 0.05) were observed between F2 and F3. This led to high values of the STC RI (Table 7).

Persimmon flour contains other compounds such as protein and dietary fiber. These macromolecules can interact with polyphenols, which would be physically trapped in the persimmon flour matrix. Therefore, phenolic acids could be liberated from their linkage to fiber and protein during in vitro digestion, yielding soluble phenolic acids [43]. Furthermore, a higher solubilization of these compounds during the gastric and duodenal steps has been observed by other authors in carob flour [43].

Lucas-González et al. [44] formulated spaghetti with 3% and 6% persimmon flour obtained from the Rojo Brillante and Triumph varieties. The bioaccessibility and antioxidant capacity of extractable polyphenols and non-extractable polyphenols after simulated in vitro digestion were determined. Two new compounds were found in spaghetti enriched with persimmon flour, gallic acid, and p-coumaric-o-hexoside, altering the polyphenolic profile and increasing the antioxidant capacity. Pico et al. [45] reported a significant increase in soluble phenolic content after digestion of cakes where 10% wheat flour was replaced by banana flour. They attributed it not only to a higher phenolic content in banana-added cakes but also to a higher phenolic bioaccessibility.

Regarding the FRAP and DPPH values of the soluble fraction after in vitro digestion, muffins enriched with persimmon flour led to a significantly (*p <* 0.05) higher antioxidant activity in the soluble fraction compared to the control muffin (Table 7). F2 and F3 indicated the highest antioxidant activity and no significant differences (*p >* 0.05) were observed between them. Hsu et al. [46] suggested that the high antioxidant capacity may be due to the partial hydrolysis of hydrolysable polyphenolic compounds by enzymes during the digestion process.

F3 had the highest TCC and F0 had the lowest (*p <* 0.05) TCC, which increased with the percentage of persimmon flour added to the muffin. A similar trend was observed in the antioxidant activity measured by the FRAP and DPPH methods in carotenoid extract. After in vitro digestion, the carotenoid content, which undergoes intestinal micellization and absorption, decreased, obtaining an RI below 100% (Table 7). The highest RI was obtained for F1 where the values decreased as the percentage of persimmon flour increased in the muffins.

The decrease in TCC content during in vitro digestion could be attributed to its lipophilic nature and to the matrix effect. In fact, the bioavailability of carotenoids present in food of plant origin is often lower than carotenoids present in food not of plant origin, probably because they are trapped by plant membranes and dietary fibers [47]. Our results agree with previous studies with plant foods where carotenoids showed a relatively high stability with, on average, >80% remaining after in vitro digestion [48].

## 4. Conclusions

Persimmon is naturally bestowed with bioactive molecules, including polyphenolic compounds, mainly tannins, and carotenoids. However, astringent varieties are often discarded and cannot be sold, due to their characteristic astringency. In this work, an astringent persimmon flour is used to obtain a non-astringent food product, avoiding the application of a de-astringency treatment, which involves an extra cost. Gluten-free muffins with persimmon flour up to 30%, and good texture, structural, and sensory properties are obtained, and the astringency is removed during muffin baking due to tannins’ insolubilization. The use of persimmon flour enriches the muffins with bioactive compounds with high antioxidant activity and high recovery index values for tannins and carotenoids after in vitro digestion. This study can contribute to the dietary diversity of patients with celiac disease by providing muffins with a better nutritional profile, regarding bioactive compounds with antioxidant properties.

## Figures and Tables

**Figure 1 foods-11-03357-f001:**
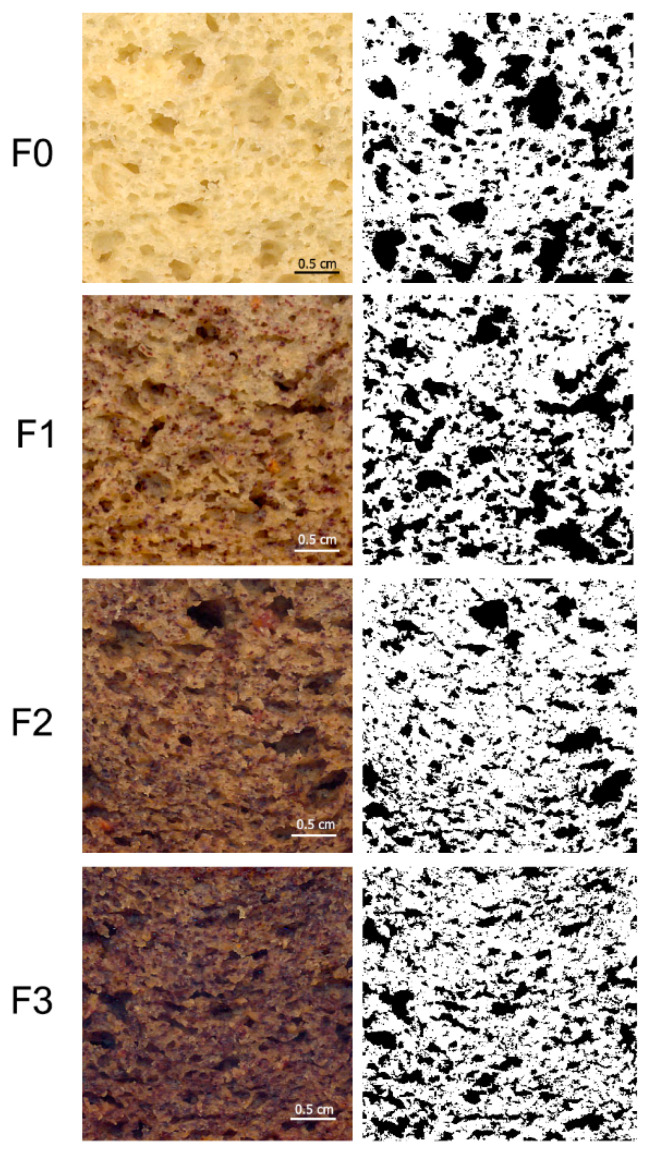
Scanned crumbs (left) and binarized images of scanned crumbs (right) of the muffins. F0: muffins without persimmon flour (control sample). F1: 10% replacement of corn flour with persimmon flour. F2: 20% replacement of corn flour with persimmon flour. F3: 30% replacement of corn flour with persimmon flour.

**Figure 2 foods-11-03357-f002:**
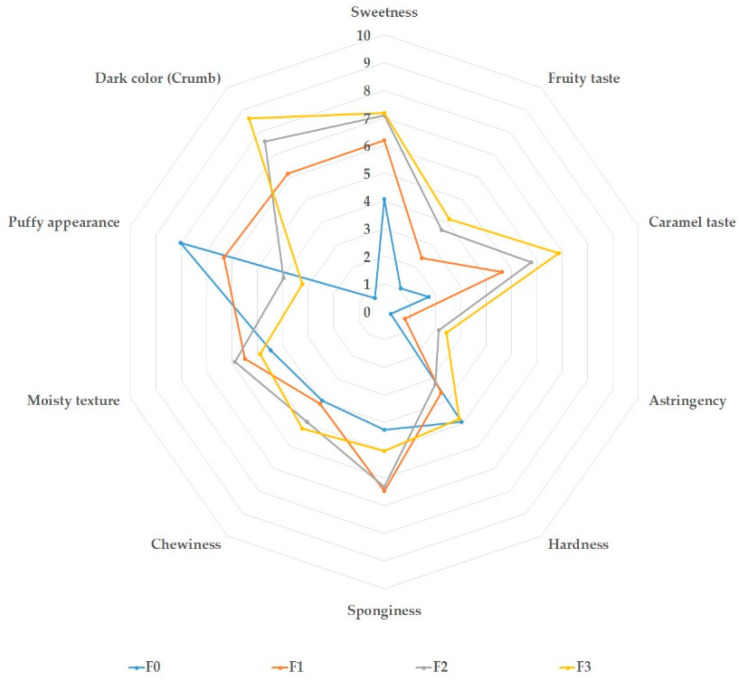
Mean QDA attributes scores performed by a trained panel (*n* = 10). Minimum score = 0 and maximum score = 10. F0: muffins without persimmon flour (control sample). F1: 10% replacement of corn flour with persimmon flour. F2: 20% replacement of corn flour with persimmon flour. F3: 30% replacement of corn flour with persimmon flour.

**Table 1 foods-11-03357-t001:** Formulations of the muffin batters (g).

Ingredients	F0	F1	F2	F3
Corn flour	160	144	128	112
Persimmon flour	0	16	32	48
Sunflower oil	110	110	110	110
Egg	100	100	100	100
Sugar	130	130	130	130
Baking powder	6	6	6	6
Xanthan gum	1	1	1	1
Water	44	44	44	44

F0: muffins without persimmon flour (control sample). F1: 10% replacement of corn flour with persimmon flour. F2: 20% replacement of corn flour with persimmon flour. F3: 30% replacement of corn flour with persimmon flour.

**Table 2 foods-11-03357-t002:** Crumb and crust color attributes (L*, h*, and C*) and total color differences (ΔE*) of the muffins.

Samples	L*	h*	C*	ΔE*
Crumb				
F0	70.28 ^d^ ± 1.77	94.49 ^d^ ± 1.23	26.88 ^c^ ± 1.46	-
F1	44.97 ^c^ ± 1.38	81.20 ^c^ ± 0.83	23.15 ^b^ ± 0.75	26.30 ^a^ ± 1.06
F2	31.95 ^b^ ± 1.29	70.98 ^b^ ± 2.08	19.69 ^a^ ± 1.13	40.18 ^b^ ± 0.71
F3	28.93 ^a^ ± 0.68	65.75 ^a^ ± 1.23	18.65 ^a^ ± 1.23	44.07 ^b^ ± 0.85
Crust				
F0	66.20 ^c^ ± 4.13	82.89 ^c^ ± 3.79	34.53 ^b^ ± 1.29	-
F1	48.61 ^b^ ± 0.56	69.94 ^b^ ± 3.57	27.29 ^a^ ± 1.48	20.91 ^a^ ± 2.12
F2	41.12 ^a^ ± 2.07	69.34 ^b^ ± 1.50	26.34 ^a^ ± 1.83	26.26 ^a^ ± 1.54
F3	37.23 ^a^ ± 1.50	65.25 ^a^ ± 1.64	27.63 ^a^ ± 1.41	28.34 ^a^ ± 0.61

F0: muffins without persimmon flour (control sample). F1: 10% replacement of corn flour with persimmon flour. F2: 20% replacement of corn flour with persimmon flour. F3: 30% replacement of corn flour with persimmon flour. The mean values in a column with different superscript letters differ significantly (*p <* 0.05) according to ANOVA (LSD multiple range test).

**Table 3 foods-11-03357-t003:** Muffin height and crumb structure parameters.

Samples	Air Cell Count	Total Cell Area (cm^2^)	Air Cell Size (cm^2^)	Height (cm)
F0	568.71 ^a^ ± 80.12	2.08 ^a^ ± 0.34	0.0036 ^a^ ± 0.0005	4.80 ^a^ ± 0.18
F1	519.33 ^a^ ± 25.39	2.11 ^a^ ± 0.35	0.0041 ^a^ ± 0.0007	3.82 ^b^ ± 0.22
F2	952.16 ^b^ ± 136.40	1.84 ^a^ ± 0.45	0.0020 ^b^ ± 0.0007	3.81 ^b^ ± 0.32
F3	1212.83 ^c^ ± 115.85	1.95 ^a^ ± 0.09	0.0016 ^b^ ± 0.0001	3.46 ^b^ ± 0.14

F0: muffins without persimmon flour (control sample). F1: 10% replacement of corn flour with persimmon flour. F2: 20% replacement of corn flour with persimmon flour. F3: 30% replacement of corn flour with persimmon flour. The mean values in a column with different superscript letters differ significantly (*p <* 0.05) according to ANOVA (LSD multiple range test).

**Table 4 foods-11-03357-t004:** TPA parameter values obtained in the muffins.

Samples	Hardness (N)	Springiness	Cohesiveness	Chewiness (N)	Resilience
F0	1.33 ^b^ ± 0.24	0.9 ^a^ ± 0.08	0.75 ^a^ ± 0.06	0.90 ^b^ ± 0.18	0.37 ^a^ ± 0.07
F1	0.95 ^a^ ± 0.21	0.92 ^a^ ± 0.03	0.77 ^ab^ ± 0.04	0.67 ^a^ ± 0.14	0.38 ^a^ ± 0.04
F2	1.17 ^b^ ± 0.27	0.91 ^a^ ± 0.04	0.78 ^ab^ ± 0.04	0.82 ^b^ ± 0.15	0.40 ^ab^ ± 0.06
F3	1.16 ^b^ ± 0.2	0.94 ^a^ ± 0.03	0.80 ^b^ ± 0.01	0.87 ^b^ ± 0.16	0.43 ^b^ ± 0.02

F0: muffins without persimmon flour (control sample). F1: 10% replacement of corn flour with persimmon flour. F2: 20% replacement of corn flour with persimmon flour. F3: 30% replacement of corn flour with persimmon flour. The mean values in a column with different superscript letters differ significantly (*p <* 0.05) according to ANOVA (LSD multiple range test).

**Table 5 foods-11-03357-t005:** Soluble tannin content (STC) and antioxidant activity of muffins.

Samples	STC (g GAE/100 g)	FRAP [Trolox] (µmol/g)	DPPH [Trolox] (µmol/g)
F0	-	0.52 ^a^ ± 0.01	-
F1	0.006 ^a^ ± 0.003	1.44 ^b^ ± 0.11	0.03 ^b^ ± 0.10
F2	0.023 ^b^ ± 0.006	3.96 ^c^ ± 0.21	0.32 ^c^ ± 0.03
F3	0.036 ^c^ ± 0.006	4.79 ^d^ ± 0.25	1.10 ^d^ ± 0.03

F0: muffins without persimmon flour (control sample). F1: 10% replacement of corn flour with persimmon flour. F2: 20% replacement of corn flour with persimmon flour. F3: 30% replacement of corn flour with persimmon flour. Means values in a column with different superscript letters differ significantly (*p <* 0.05) according to ANOVA (LSD multiple range test).

**Table 6 foods-11-03357-t006:** Total carotenoid content (TCC) and antioxidant activity of muffins.

Samples	TCC at 450 nm (mg β-Carotene/100 g)	FRAP Trolox (µmol/g)	DPPH Trolox (µmol/g)
F0	6.50 ^a^ ± 1.75	-	0.83 ^a^ ± 0.06
F1	10.64 ^b^ ± 1.35	-	0.91 ^a^ ± 0.05
F2	15.74 ^c^ ± 1.77	2.65 ^b^ ± 0.21	1.46 ^b^ ± 0.06
F3	17.76 ^c^ ± 1.83	3.25 ^c^ ± 0.39	1.51 ^b^ ± 0.06

F0: muffins without persimmon flour (control sample). F1: 10% replacement of corn flour with persimmon flour. F2: 20% replacement of corn flour with persimmon flour. F3: 30% replacement of corn flour with persimmon flour. Means values in a column with different superscript letters differ significantly (*p <* 0.05) according to ANOVA (LSD multiple range test).

**Table 7 foods-11-03357-t007:** Soluble tannin content (STC), antioxidant activity of the soluble fraction (FRAP-s and DPPH-s) and of the carotenoids fraction (FRAP-c and DPPH-c) after in vitro digestion, and RI% of STC and carotenoids of the muffins.

After In Vitro Digestion	F0	F1	F2	F3
STC (g GAE/100 g)	0.088 ^a^ ± 0.002	0.115 ^b^ ± 0.002	0.166 ^c^ ± 0.001	0.179 ^c^ ± 0.016
FRAP-s (µmol Trolox/g)	0.074 ^a^ ± 0.011	12.384 ^b^ ± 0.065	13.55 ^c^ ± 0.008	14.65 ^c^ ± 0.049
DPPH-s (µmol Trolox/g)	2.176 ^a^ ± 0.007	2.623 ^b^ ± 0.003	2.807 ^c^ ± 0.035	2.839 ^c^ ± 0.060
RI% STC	-	2023.56	730.98	502.6
TCC 450 nm (mg β-carotene/100 g)	0.07 ^a^ ± 0.74	8.73 ^b^ ± 0.37	9.72 ^bc^ ± 0.05	10.90 ^c^ ± 0.46
FRAP (µmol Trolox/g)	-	-	0.20 ^c^ ± 0.03	0.41 ^d^ ± 0.07
DPPH (µmol Trolox/g)	2.71 ^a^ ± 0.02	2.97 ^b^ ± 0.01	3.29 ^c^ ± 0.04	3.58 ^d^ ± 0.12
RI% Carotenoids 450 nm	1.14	82.08	61.75	61.37

F0: muffins without persimmon flour (control sample). F1: 10% replacement of corn flour with persimmon flour. F2: 20% replacement of corn flour with persimmon flour. F3: 30% replacement of corn flour with persimmon flour. Means values in a row with different superscript letters differ significantly (*p <* 0.05) according to ANOVA (LSD multiple range test).

## Data Availability

The data used to support the findings of this study can be made available by the corresponding author upon request.

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
