# Peer review of "Evaluation of the Bioactive Compounds, and Physicochemical and Sensory Properties of Gluten-Free Muffins Enriched with Persimmon ‘Rojo Brillante’ Flour"

_foods, 2022, doi:10.3390/foods11213357_

Round 1

Reviewer 1 Report

In general, this is an interesting article that covers the overall effect of flour addition on the quality of muffins.

In my opinion, the addition was relatively low to have a significant effect on the number of bioactive compounds, physicochemical, and sensory properties of muffins. However, the results show that it has an effect so please explain how and why.

Line 4, at the end of a title, remove a dot “.”

Since the Authors investigated the effect of partial replacement of corn flour with persimmon flour, it would be good to know the chemical composition of both flours. In my opinion, such information should be provided in the Materials chapter. It would be helpful to explain changes in the physicochemical parameters of muffins. F.ex. I assume it is rich in sugars which led to the brewing effect.

Conclusions: “providing muffins with a better nutritional profile”. I think the conclusions should be more specific.

Author Response

In general, this is an interesting article that covers the overall effect of flour addition on the quality of muffins.

In my opinion, the addition was relatively low to have a significant effect on the number of bioactive compounds, physicochemical, and sensory properties of muffins. However, the results show that it has an effect so please explain how and why.

We think the amount of persimmon added to the muffins is significant because our persimmon flour has a very low water content (around 2%), so the bioactive compounds are concentrated. That is why an effect is observed in the results.

Line 4, at the end of a title, remove a dot “.”

Thanks for noticing. The dot has been removed.

Since the Authors investigated the effect of partial replacement of corn flour with persimmon flour, it would be good to know the chemical composition of both flours. In my opinion, such information should be provided in the Materials chapter. It would be helpful to explain changes in the physicochemical parameters of muffins. F.ex. I assume it is rich in sugars which led to the brewing effect.

The chemical composition of both flours has been included in M&M section (lines 93 and 97)

Conclusions: “providing muffins with a better nutritional profile”. I think the conclusions should be more specific.

The conclusion section has been rewritten.

Reviewer 2 Report

Line 9 changes to “celiac disease”

Line 10 changes to “the properties”

Line 77-80 In many studies (Chen et al. 2008 “Radical Scavenging Activity and Phenolic Compounds in Persimmon (Diospyros kaki L. cv. Mopan)”, Karaman et al. 2014 “Bioactive and physicochemical properties of persimmon as affected by drying methods”, Cho et al. 2003 “Preparation of yogurt supplemented with sweet persimmon powder and quality characteristics”  in the literature about persimmon flour or dried powder, the total phenolic content and carotenoids have been determined to be extremely high compared to the value you find. What is the reason for this?

Why is there no standard deviation in the ∆E values in Table 2?

Line 261 What is the amount of HMF that increases with the increase of the Maillard reaction? Is it within limits?

In Table 4, where the texture properties are given, the hardness value of the F1 sample was decreased significantly, but this was not observed in muffins with other additions. What is the reason for this?

All the values in Table 5 are much lower than the literature. What is the reason for this?

Reviewer 3 Report

The manuscript is written with clear understanding of the project addressed. However, there are some concerns that need to be addressed to enhance the quality of the manuscript. My specific comments are as follows:

Abstract:

Add the main finding of your study.

Introduction:

“Many studies have been conducted in the past on the development of bakery products containing dietary…” Explain those studies

“Celiac disease is an autoimmune disorder that primarily affects the small intestine caused by…” Add citation

“In this regard, persimmon varieties can be divided into two categories according to astringency: astringent…” Add citation

Based on your objectives, please compare how your study is different from those that have already been published

Results and discussion:

Fluctuation occurs for C*. justify

“…attributed to the intensification of non-enzymatic browning (caramelization and Maillard reaction) that..” any reference?

Elaborate more on figure 1

“Therefore, no significant differences (p > 0.05) in total cell area were found among the samples.” Give justification

“Despite the differences observed in muffin height and crumb structure, similar mechanical parameters were obtained in all muffins.” No significant differences obtained for hardness and springiness. Discuss mean comparison of other TPA parameter too

“Other researchers have addressed the effect of using persimmon flour in the sensory properties of muffins.” Explain those studies

“Therefore, phenolic acids could be liberated from their linkage to fiber and protein during in vitro digestion, yielding soluble phenolic acids.” Add citation

The findings lack in terms of justification and major findings.

Conclusions:

Emphasize more on the main finding.

General comments:

Please check the reference styles and grammar of the manuscript.

Author Response

The manuscript is written with clear understanding of the project addressed. However, there are some concerns that need to be addressed to enhance the quality of the manuscript. My specific comments are as follows:

Abstract:

Add the main finding of your study.

The following sentence has been added to the abstract: “Astringent varieties can be used, without  applying a de-astringency treatment, as the astringency is removed during muffin baking due to tannins’ insolubilization” (lines 22-24).We consider it an important finding.

Introduction:

“Many studies have been conducted in the past on the development of bakery products containing dietary…” Explain those studies

The explanation has been added (lines 33-39).

“Celiac disease is an autoimmune disorder that primarily affects the small intestine caused by…” Add citation

Thanks for noticing. The citation has been added (line 47).

“In this regard, persimmon varieties can be divided into two categories according to astringency: astringent…” Add citation

Thanks for noticing. The citation has been added (line 66).

Based on your objectives, please compare how your study is different from those that have already been published

The use of “Rojo Brillante” variety, which is often discarded due to its astringency, to prepare non-astringent food ( muffins) is the main difference with other studies.

Results and discussion:

Fluctuation occurs for C*. justify “…attributed to the intensification of non-enzymatic browning (caramelization and Maillard reaction) that..” any reference?

Thanks for noticing. The reference has been added (line 272).

Elaborate more on figure 1

More information about figure 1 has been  added ( lines 278-279)

“Therefore, no significant differences (p > 0.05) in total cell area were found among the samples.” Give justification

A justification has been  added ( lines 291-293)

“Despite the differences observed in muffin height and crumb structure, similar mechanical parameters were obtained in all muffins.” No significant differences obtained for hardness and springiness. Discuss mean comparison of other TPA parameter too

Chewiness, cohesiveness, and resilience are discussed in lines 319-321

“Other researchers have addressed the effect of using persimmon flour in the sensory properties of muffins.” Explain those studies

We have deleted the word “flour” in line 352 since the studies carried out by Yesilkanat and Savlak, and Abdallah et al. were performed with persimmon powder and puree in bakery products

“Therefore, phenolic acids could be liberated from their linkage to fiber and protein during in vitro digestion, yielding soluble phenolic acids.” Add citation

Thanks for noticing. The citation has been added (line 436)

The findings lack in terms of justification and major findings.

The main findings have been included in the abstract (lines 22-24) and conclusion section (lines 472-474)

Conclusions:

Emphasize more on the main finding.

Conclusion section has been reworded emphasizing the main findings

General comments:

Please check the reference styles and grammar of the manuscript.

Reference style has been checked and grammar has been revised by a native  English speaker.

Round 2

Reviewer 1 Report

The manuscript had been improved and can be published. 

Author Response

Thank you very much for your comment 

Reviewer 2 Report

The innovation of the research in the Introduction should be highlighted. Reveal the originality of the study.

Emphasize more on the main finding.

the differences in the results in the result and discussion section have not been adequately explained.

Author Response

The innovation of the research in the Introduction should be highlighted. Reveal the originality of the study.

The introduction section has been improved explaining the main novelty of this work (lines: 64-65; 67-72; 80-82).

Emphasize more on the main finding.

The main findings have been emphasized (lines 267-269; 298-300, 400-405).

the differences in the results in the result and discussion section have not been adequately explained.

The differences in the results have been better explained (Lines 185-187; 190-194; 248-252; 262-263).